# Technical Report on the 4th Edition of Knowledge Graph Construction Workshop Challenge*

Sitt Min Oo[1,*,†], Enrique Iglesias[2,3,*,†], Jakub Duchateau[4], Marco Grassi[5], Pano Maria[6], Els de Vleeschauwer[1], Adrián Martínez-Balea[7], Mario Scrocca[5] and Maria-Esther Vidal[2,3,8]

[1]*IDLab, Department of Electronics and Information Systems, Ghent University*

[2]*L3S Research Center, Hannover, Germany*

[3]*Leibniz University of Hannover, Hannover, Germany*

[4]*Montefiore Institute, University of Liège, Belgium*

[5]*Cefriel – Politecnico di Milano, Milan, Italy*

[6]*ModelDesk, Schiedam, Netherlands*

[7]*CiTIUS, Universidade de Santiago de Compostela, Spain*

[8]*TIB Leibniz Information Center for Science and Technology, Hannover, Germany*

## Abstract

Knowledge graph (KG) construction has become an important research area, motivating the development of declarative mapping languages such as the RDF Mapping Language (RML). The fourth edition of the Knowledge Graph Construction Challenge, organized by the Knowledge Graph Construction Workshop (KGCW), evaluates the current state of KG creation engines through three complementary tracks: conformance, performance, and mapping methodology. The Conformance track assesses compliance with the latest RML specification across five modules that cover core RML functionality and its extensions. The Performance track evaluates execution time and memory consumption under different KG construction scenarios, including varying input sizes, duplicate rates, and join characteristics. The Mapping Methodology track encourages the exploration of alternative approaches to KG creation beyond traditional RML-based solutions. A total of five engines participated in the conformance track and three in the performance track. This paper presents the challenge design, participant submissions, and results across the different tracks, highlighting the current capabilities of the engines.

## Keywords

Knowledge Graphs, RDF Mapping Language

# 1. Introduction

Knowledge graphs (KGs) have become increasingly widespread in recent years, driven by the rapid growth in the volume of daily data. Major companies such as Google, Netflix, Amazon, and Microsoft use KGs to establish relationships among products, concepts, and topics in order to enhance the user experience [1]. As a result, numerous methods and mapping languages have been developed to support KG creation. One such language is the RDF Mapping Language (RML) [2]. RML is a mapping language used in the domain of declarative KG construction. It has seen significant updates to its specifications [3], making the language modular[1] and extending the capabilities of compliant mapping engines to support

---

*Dubrovnik'26: Seventh International Workshop on Knowledge Graph Construction, May 10, 2026, Dubrovnik, HR*

*You can use this document as the template for preparing your publication. We recommend using the latest version of the ceurart style.

*Corresponding author.

†These authors contributed equally.

✉ x.sittminoo@ugent.be (S. M. Oo); iglesias@l3s.de (E. Iglesias); Jakub.Duchateau@uliege.be (J. Duchateau); marco.grassi@cefriel.com (M. Grassi); pano@modeldesk.io (P. Maria); els.devleeschauwer@ugent.be (E. de Vleeschauwer); adrianmartinez.balea@usc.es (A. Martínez-Balea); mario.scrocca@cefriel.com (M. Scrocca); maria.vidal@tib.eu (M. Vidal)

🄳 0000-0001-9157-7507 (S. M. Oo); 0000-0002-8734-3123 (E. Iglesias); 0009-0009-5090-8192 (J. Duchateau); 000-0003-3139-3049 (M. Grassi); 0009-0000-2598-1894 (P. Maria); 0000-0002-8630-3947 (E. de Vleeschauwer); 0009-0005-2185-400X (A. Martínez-Balea); 0000-0002-8235-7331 (M. Scrocca); 0000-0003-1160-8727 (M. Vidal)

[1]https://w3id.org/rml/portal

new features of KG construction techniques (e.g., enabling KG construction from nested heterogeneous data formats using RML-LV [4]).

Such drastic changes in the specifications necessitate new test cases to verify the conformance of state-of-the-art RML mapping engines with the latest specifications. Moreover, supporting new features in KG construction may affect the performance of mapping engines in terms of resource usage and execution time. Thus, there is a need for an extensive benchmark evaluation of different engines to better inform users when choosing a mapping engine for their use cases, particularly regarding the trade-off between full specification compliance and performance.

Previous editions of the challenge have focused evaluating mapping engines based on RML and its ecosystems, despite the significant potential in investigating alternative approaches to KG construction outside of RML. Therefore, a challenge scenario is required to investigate such alternative approaches to KG construction. The focus of the challenge is to evaluate the novelty of each participant's methods for defining and executing mappings, while also assessing the practical benefits, such as the (re-)usability and the expressiveness of their mapping model for KG construction.

For this paper, we organize the 4th Edition of KGCW Challenge where each participants submit their implementations (Section 2) to one of the three different tracks: i) Track 1 evaluates mapping engines for conformance against the selected RML modules' specifications (Section 3.1), ii) Track 2 measures the resource usage and execution time of mapping engines under diverse scenarios (Section 3.2), and iii) Track 3 explores novel techniques on KG construction form heterogeneous and (semi-)structured data beyond RML-based approaches (Section 3.3). We present comparative insights into participants' results (Section 3) to determine the best solutions for each challenge track.

## 2. Engines

### 2.1. CARML

CARML is a Java library that transforms structured and semi-structured data (CSV, JSON, and XML documents, as well as relational databases such as MySQL and PostgreSQL) into RDF according to RML mapping rules, and it supports the latest RML specification [3]. It is built on the Eclipse RDF4J toolkit and can emit RDF either natively through RDF4J or, through a converter, via Apache Jena. RML mappings are parsed into CARML's domain model by an annotation-driven RDF-to-object mapper, and new source formats are contributed as service-loaded resolver plugins, so the set of supported formats can be extended without modifying the engine core. Throughout the mapping pipeline, records are processed as reactive streams using Project Reactor[2].

For the 2026 challenge, CARML's execution model is organized around a single unified abstraction: the *logical view* [4]. Every mapping is resolved into a graph of logical views before execution. A mapping that declares an explicit *rml:LogicalView* (RML-LV) is used directly, whereas a mapping that uses only a bare *rml:LogicalSource* (RML-Core) is wrapped into an implicit logical view over the same source. As a result, RML-Core and RML-LV mappings traverse exactly the same execution pipeline, and the engine's optimizations apply uniformly to both. Beyond its source binding, a logical view carries the structural annotations available from the specification: key, non-null, and uniqueness constraints, value multiplicity, and dependencies between referenced fields. CARML uses these annotations to plan execution: trimming each source projection to the fields a triples map actually references, skipping deduplication where constraints already guarantee uniqueness, ordering joins, and decomposing a view into independently evaluable parts.

The evaluation of a logical view is delegated to one of two interchangeable backends. The *reactive* backend evaluates a view as a chain of Project Reactor stream operators, processing the source record by record and holding intermediate state on the JVM heap; it is designed for bounded, predictable memory usage. The *in-process database* backend stages each source into an embedded DuckDB instance and evaluates the view as SQL, using vectorized execution for joins and aggregations and spilling

---

[2] https://projectreactor.io

intermediate state to disk when the working set exceeds a configured memory limit. The backend is selected automatically per view, or forced for all views from the command line. Because both backends consume the same logical-view plan and produce identical RDF, CARML is reported as two entries in this challenge—*CARML* (in-process database) and *CARML-reactive* (reactive)—to characterize the trade-off between the two execution strategies within a single engine.

CARML passes all test cases across the five Track 1 modules: RML-Core, RML-IO, RML-CC, RML-FNML, and RML-LV. For the Track 2 performance scenarios, executing the largest cases was constrained by the benchmark VM's available disk: the highest-volume cells produce several gigabytes of N-Triples per run, and across the five required runs, their accumulated output can exceed the free disk space, which we mitigated by reclaiming each run's output before starting the next. A further group of cells was excluded in advance because a single run's output would not fit on the benchmark VM. These cells use the RML-CC gather construct to collect a cartesian product (the many-to-many join of a million-row source, or the product of two value arrays per record) into an RDF list; since every list member additionally expands into an *rdf:first*/*rdf:rest* chain, the output grows quadratically with the input. Extrapolating from smaller runs, a single such cell would produce from tens of terabytes to petabytes of RDF triples, far exceeding the VM's available disk. The limitation is output volume rather than the engine: CARML streams these gathers into bounded memory, so, given sufficient disk space and time, it would, in principle, complete the cells. Cells in either category are reported as did-not-finish.

The version of CARML used in this work is the development build on the rml-cg branch, publicly available on GitHub[3].

## 2.2. SDM-RDFizer

SDM-RDFizer [5, 6] is a KG creation engine capable of transforming structured (i.e., CSV and relational databases) and semi-structured data (i.e., JSON and XML) into RDF triples, following defined mapping rules. It supports the latest RML specification.

SDM-RDFizer adopts a two-fold approach to KG creation, consisting of two main modules: **Triples Maps Planning** (TMP) and **Triples Maps Execution** (TME). Each module plays a distinct role in the KG creation process and employs a set of data structures and operators to handle various aspects such as join execution and duplicate removal. TMP determines the execution order of RML Triples Maps to minimize memory usage. TME then generates the KG according to the execution plan given by TMP. To interpret RML triples maps, SDM-RDFizer uses a SPARQL-based parser. This parser employs four SPARQL queries: the main query extracts core mapping information, such as the logical source, *rml:sub-jectMap*, *rml:predicateObjectMap*, join conditions, logical dumps, and function calls. Meanwhile, the remaining queries handle nested structures, such as collections, functions, and fields, in logical views. To support the transformation of different types of triples maps, SDM-RDFizer implements several specialized operators. The **Simple Object Map** (SOM) operator executes *rml:template* and *rml:reference*; the **Object Reference Map** (ORM) operator handles parent triples maps; and the **Object Join Map** (OJM) operator executes joins. For duplicate detection, SDM-RDFizer uses hash tables known as **Predicate Tuple Tables** (PTTs). Each generated triple is compared against the corresponding PTT; if it already exists, it is discarded as a duplicate. Otherwise, the triple is added to both the PTT and the KG. For joins, SDM-RDFizer caches the results in a structure called the **Predicate Join Tuple Table** (PJTT) to avoid redundant join computations.

For the 2024 challenge [7] and 2025 challenge [8], SDM-RDFizer was extended to support the latest RML modules, including RML-FNML, RML-Star, RML-IO, RML-CC, and RML-LV. In the case of RML-IO, SDM-RDFizer was enhanced to accept new input types, including compressed files (e.g., ZIP, TAR), SPARQL endpoints, and remote data sources. It also gained the ability to compress the generated KG into various formats, split triples into multiple files, and export in different RDF serializations (e.g., JSON-LD, RDF/XML, Turtle).

To support RML-FNML, a specialized operator was introduced to execute functions during data transformation, incorporating strategies from FunMap [9]. FunMap is a triples map translator that replaces

---

[3]https://github.com/carml/carml/tree/rml-cg

triple maps that contain functions with equivalent triple maps by executing those functions and transforming the input data accordingly. SDM-RDFizer applies the same principle, enabling on-the-fly data transformation by executing functions.

For RML-Star, a specialized operator was developed to handle the *rml:quotedTriplesMap* construct introduced in this module. Additionally, PJTT was extended to store joins that may occur in either the *rml:subjectMap* or the *rml:objectMap*.

For RML-CC, an operator was implemented to gather the values indicated in *rml:gather*, organize the values in the container type defined in *rml:gatherAs*, and finally generate all the corresponding triples.

For RML-LV, an operator was introduced to preprocess data sources according to the logical view. For this purpose, this operator can project data sources, join them, and process sources with nested data in a different format, such as a CSV file containing JSON data.

For the 2026 Challenge, SDM-RDFizer was further enhanced to incorporate the refinements introduced in the evolving RML modules. The version of SDM-RDFizer used in this work is 4.7.5.14. SDM-RDFizer is publicly available on GitHub [4].

The results of the challenge highlighted several directions for future development. The operator responsible for RML-CC processing will be optimized to handle large data sources more effectively. The RML-LV operator will be extended to support additional input formats and combinations of heterogeneous data sources beyond JSON and CSV. An improved RML-Star operator is planned for future releases. Finally, the current query-based mapping parser will be replaced by a dedicated parser to reduce complexity and improve maintainability.

## 2.3. BURP

BURP [10], which stands for *Basic and Unassuming RML Processor*, was initially developed to validate RML-CC test cases and the conformance infrastructure, with an RML processor that supports only the new specification. BURP is a naive implementation of RML, initially developed from scratch in Java and adopted Jena. The implementation uses simple algorithms without optimizations, hence *basic*. BURP was also useful when developing and testing the 2024 test infrastructure, as most RML engines at that time supported combinations of R2RML, the "old" RML, and new RML. As a consequence, some test cases did not comply with the new specifications but yielded correct results from those engines. BURP is unassuming in that it was developed with test cases and sanity-checked against the documentation provided in the specifications. Over time, parts of BURP were rewritten in Kotlin.

With this year's submission, BURP has seen an increase in conformance compared to 2024 Conformance [11]. This was thanks to the implementation of RML-LV, and in BURP-Error, a fork of BURP, RML-IO targets were implemented, and the main contribution of that fork is RER [12], a RML Execution Report that describes the errors of the run according to the RER taxonomy[5] of errors. In addition to the usual outputs, it adds an RDF report. As BURP-Error is mainly used as a command-line interface (CLI), it also provides a textual representation of the report, with snippets of the mapping where the error is highlighted in the turtle mapping. In BURP-Error, the main source of non-compliance is still RML-IO, mainly due to issues in some test cases (kg-construct/rml-io#151 and kg-construct/rml-io#167), the unavailability of UTF-16, and issues with datatype conversions from SPARQL. With BURP-Error, they did some refactoring in BURP. We replace the Jena RDF model used for RDF generation with the authors' own RDF data model implementation, enabling them to generate unsafe RDF as required by RML and to more easily post-process collections and containers. Also, all data structures that represent the mapping execution plan in BURP-Error have a common ancestor `PlanNode` that enables generic operations and represents the operations as a tree, which enables generic operations on nodes, or getting ancestors or descendants of a plan node, which enables a scope system, for example, for base IRI or in join conditions. The Java Service Provider Interface (SPI) was used to provide two interfaces to enable extending the supported functions and logical sources. By replacing the JSONPath implementation

---

[4]https://github.com/SDM-TIB/SDM-RDFizer
[5]https://w3id.org/dre/rer

with an RFC 9535 [13] compliant library[6], it contributed to detecting 22 test cases using non-compliant JSONPath reference formulations.

BURP and BURP-Error are both available under the MIT License on GitHub. The RML-LV implementation can be found in BURP in the kgc-2025-conformance branch[7] or in BURP-Error[8] for all the other changes, including RML-LV.

Even though some test cases failed, the community adopts different approaches to reporting them (e.g., omitting the test case vs. reporting the failure but not the issue with the test case). One possible approach would be to enrich the reporting options with additional statuses for a failing test case when the failure is due to an issue in the test case itself, according to the implementers.

With the introduction of the RER report and taxonomy of errors, the question arises: What is an error beyond just a flag, an exit code, or empty output datasets? Should test cases expecting an error also specify the kind of error to output? Beyond the error type, if the RER report is adopted by other RML processors, which parts of that report should match remains an open question.

## 2.4. MappingWeaver with RMLViewer

MappingWeaver[9] is a mapping engine implemented based on the language-agnostic algebra for KG construction proposed by Min Oo and Hartig [14]. The algebra captures mapping definitions independently of the mapping language being used. MappingWeaver is an improvement of the algebra-based mapping engine, RMLWeaver-js[10], which was presented the 3rd edition of KGCW challenge [15].

Similar to RMLWeaver-js, MappingWeaver has the same end-to-end execution flow consisting of two stages: i) *translation* where the input RML mapping document is translated into a mapping plan consisting of algebraic operators (MappingLoom-rs[11]), and ii) *execution*, where the generated mapping plan is executed for KG construction (MappingWeaver). The two stages are implemented separately to enable reuse of the translation step in different contexts (i.e., different execution engine implementations can use the same mapping plan generated by the translation step).

For the translation stage, MappingWeaver implements the original translation algorithm from Min Oo and Hartig [14], with extensions to support the latest RML modules. Since RML-Core is backwards compatible with RML v1.1.2[12], only minimal changes are needed to support the RML-Core module. RML-IO is supported partially due to the migration of RML logical source related IRIs of RML v1.1.2 to RML-IO[13] without the latest features of RML-IO. RML-FNML is supported since the semantics of data transformation through functions can be captured by introducing additional *extension functions* according to the algebra formalization [14]. On the other hand, fully supporting modules such as RML-CC, RML-Star, and RML-LV require substantial changes and extensions to the underlying algebra formalization to support their features. However, by using RMLViewer[14] to preprocess input data using RML-LV features, MappingWeaver is able to indirectly, yet fully, support RML-LV module without any changes to the underlying algebra. This preprocessing of input data happens before MappingWeaver's end-to-end execution flow; thus, RMLViewer can be used by any existing RML mapping engines to fully support the RML-LV module.

For the execution stage, unlike RMLWeaver-js, MappingWeaver is implemented as a Java application to utilize the sate-of-the-art stream processing framework Apache Flink [15]. Flink provides APIs to process bounded and unbounded data streams with ease, while dynamically scaling the applications

---

[6]https://github.com/a-sit-plus/jsonpath4k
[7]https://github.com/kg-construct/BURP/tree/kgc-2025-conformance
[8]https://github.com/jduchateau/BURP-Errors
[9]https://github.com/RMLio/MappingWeaver-java
[10]https://github.com/RMLio/rmlweaver-js
[11]https://github.com/RMLio/mappingloom-rs/
[12]https://rml.io/specs/rml/
[13]https://kg-construct.github.io/rml-resources/portal/backwards-compatibility.html
[14]https://github.com/RMLio/rmlviewer
[15]https://flink.apache.org/

under heavy workload. Consequently, implementing MappingWeaver as a Flink application enables it to leverage these features without requiring custom implementations for them.

The version of MappingWeaver used for the challenge is available on the *kgc-challenge* branch[16], and the RMLViewer is available with the version tag v2.0.0[17] on their respective GitHub repository.

## 2.5. Typhon-RML

Typhon-RML[18] is an open-source Java tool designed to support the construction of KG pipelines from heterogeneous data sources by compiling RML mappings into executable artifacts. The approach builds on a clear separation of concerns between data orchestration and mapping logic, leveraging the Chimera framework [16] to define composable semantic transformation pipelines and the Mapping Template Language (MTL) [17] to represent mapping rules. Typhon-RML generates two complementary outputs from a given RML specification: (i) a Chimera pipeline represented as an Apache Camel `route.xml` file, responsible for handling data access and integration, and (ii) a MTL (`template.vm`) file encoding mapping rules that are equivalent to the RML to be executed. This design enables flexible deployment and optimization, as data access strategies and mapping execution can be tuned independently without modifying the conceptual mapping [18]. A recent update to Typhon-RML further simplifies execution by removing the dependency on an external *Chimera Skeleton* component, allowing generated pipelines to be executed directly (i.e., via JBang and Apache Camel), thereby improving reproducibility and ease of use in evaluation settings.

The system was evaluated in the context of the KGCW 2026 challenge using Typhon-RML v0.3, relying on a custom Python test runner and RDFLib to compare produced and expected RDF graphs. Results show solid coverage of the RML-Core and RML-IO modules, with 66/76 and 63/73 tests passed, respectively. The scripts used for the evaluation and the generated artefaces for each test case are available on GitHub[19].

In addition to conformance evaluation, an exploratory effort investigated the use of an agentic approach to automatically generate mapping rules in MTL for Track 3 of the challenge. The core idea was to equip an LLM-based agent with a dedicated skill (`SKILL.md` file[20]) leveraging tools exposed through an MCP server and curated knowledge resources (MTL documentation, examples, and mapping-template documentation describing both how to write and how to execute mapping rules). The skill set included utilities to validate and inspect the generated RDF, validate RDF graphs against SHACL shapes, and compare the generated against expected RDF graphs (missing/extra triples).

## 3. KGCW 2026 Challenge Results

The KGCW 2026 Challenge [21] aims to assess the compliance of existing KG creation engines with the latest RML specification. The dataset used in this challenge is a refined version of the KGCW 2025 Challenge dataset [22]. The 2026 version of the challenge has three tracks:

- **Track 1 Compliance:** comprises the fundamental test cases that an RML engine must successfully execute to be considered compliant with the RML specification. This track is divided into five modules: RML-Core, RML-FNML, RML-IO, RML-CC, and RML-LV.

- **Track 2 Performance:** evaluates the performance of state-of-the-art KG creation engines by measuring execution time and memory consumption across a series of test cases that capture

---

[16]https://github.com/RMLio/MappingWeaver-java/tree/kgc-challenge
[17]https://github.com/RMLio/rmlviewer/releases/tag/v2.0.0
[18]https://github.com/cefriel/typhon-rml
[19]https://github.com/cefriel/typhon-rml/tree/main/evaluation/kgcw-2026
[20]https://agentskills.io/
[21]https://github.com/kg-construct/kgc-challenge
[22]https://zenodo.org/records/14970817

| Engines | RML-Core | RML-IO | RML-CC | RML-FNML | RML-LV |
|---|---|---|---|---|---|
| BURP-Error | 100.00% | 95.89% | 100.00% | 100.00% | 100.00% |
| Typhon-RML | 86.84% | 86.30% | 0.00% | 0.00% | 0.00% |
| CARML | 100.00% | 100.00% | 100.00% | 100.00% | 100.00% |
| MappingWeaver with RMLViewer | 92.10% | 28.76% | 0% | 78.94% | 100.00% |
| SDM-RDFizer | 100.00% | 100.00% | 100.00% | 100.00% | 100.00% |

**Table 1**
Test Cases of Track 1.

key factors influencing the KG creation process, including input size, the proportion of duplicate values, and the complexity of joins.

- **Track 3 Mapping Methodology:** aims to foster discussion on alternative methods for creating KGs. Its main objective is to allow participants to employ approaches beyond RML in the KG creation process.

### 3.1. Track 1: Conformance

- **RML-Core:** The 2026 release of RML-Core maintains its focus on JSON files, consistent with the 2025 version, while other formats have been reassigned to RML-IO. This iteration further introduces new test cases, including those covering child and parent maps for joins, language maps for language tags, and the subject shortcut (i.e., *rml:subject*) for subject definition[23].

- **RML-FNML:** This set comprises test cases that apply predefined functions for data transformation. The 2026 version extends this by enabling functions to be used as conditions for triple generation, as well as at the level of language and datatype maps[24].

- **RML-IO:** This set now includes a wide range of remote data sources such as endpoints, compressed files, and JSON and XML files [25]. It also defines output specifications for various formats, including Turtle, RDF/JSON, JSON-LD, and compressed formats like ZIP and TAR.

- **RML-CC:** This set includes test cases that cover collections and containers [26].

- **RML-LV:** This set features test cases where data sources are generated through projection and joins across multiple sources, including mixtures of different data formats [27]. The 2026 release includes test cases that produce errors to illustrate an invalid use of logical views. Additionally, the use of child and parent maps for joins.

For the RML-IO module, the following test cases were identified as containing errors. Even though the engines generate the correct output per the RML specification, they do not match the expected output and are therefore considered invalid.

Test case RMLSTC0006a was deemed a better fit for the RML-IO-Registry module (which is not included in the 2026 challenge) and will be ignored. [28]

Test case RMLSTC0009a is expected to raise an error because the CSV input file contains column names enclosed in quotation marks, which should be syntactically invalid. But given how CSV files work, the quotes are not considered part of the column name, thus no error is produced. To clarify this issue, a discussion was opened on GitHub.[29]

---

[23]https://kg-construct.github.io/rml-core/test-cases/docs/

[24]https://kg-construct.github.io/rml-fnml/test-cases/docs/

[25]https://kg-construct.github.io/rml-io/test-cases/docs/

[26]https://kg-construct.github.io/rml-cc/test-cases/docs/

[27]https://kg-construct.github.io/rml-lv/test-cases/docs/

[28]https://github.com/kg-construct/rml-io/issues/147

[29]https://github.com/kg-construct/rml-io/issues/150

The test cases RMLTTC0004a, RMLTTC0004c, RMLTTC0004e, RMLTTC0004f, RMLTTC0004g, RMLTTC0005b, RMLTTC0006b, RMLTTC0006c, RMLTTC0006d, and RMLTTC0006e appear to contain incorrect expected outputs and are currently under review.[30] In these cases, the expected results omit the natural datatype that should be assigned to integer values extracted from the JSON input.

CARML successfully passed all the test cases from RML-Core, RML-FNML, RML-CC, and RML-LV. For this track, CARML organizes data source processing around the concept of *logical views*. Every mapping is processed as if it has a logical view, even if it doesn't. Therefore, RML-Core and RML-LV test cases are processed thought the same execution pipeline.

For RML-IO, CARML successfully executed 61 of the 73 test cases. The failing test cases are those currently considered invalid and under review. CARML does not interpret double quotation marks as part of CSV column names and therefore generates output for this case. Additionally, CARML can generate natural data types for omitted test cases.

SDM-RDFizer successfully executed all 76 test cases in the RML-Core module. To comply with the current RML specification, its SPARQL-based parser was updated accordingly, and the join operator was extended to process constant values and evaluate templates during join execution, enabling support for child and parent maps in join conditions.

In the RML-IO module, SDM-RDFizer successfully executed 61 of the 73 test cases. The remaining cases correspond to test cases currently considered invalid and under review. In particular, SDM-RDFizer does not interpret double quotation marks as part of CSV column names and therefore generates output for this case. Likewise, when natural datatypes are omitted from the expected output, SDM-RDFizer generates literals with their corresponding natural datatypes.

For the RML-CC module, SDM-RDFizer successfully executed all 35 test cases. This module is unchanged from the 2025 challenge. Support is provided by a specialized operator that gathers the required values and generates the corresponding RDF collections or containers for the specified type.

SDM-RDFizer also successfully executed all 19 RML-FNML test cases. A dedicated operator performs function execution on the fly, ensuring that the values incorporated into the generated knowledge graph correspond to the results produced by the associated function maps.

Finally, SDM-RDFizer successfully executed all 41 RML-LV test cases. Building upon the operator introduced for the 2025 challenge, support was extended to detect invalid logical view definitions and to execute joins using child and parent maps.

BURP-Error successfully passed all the test cases from RML-Core, RML-FNML, RML-CC, and RML-LV. For this track, BURP, building upon the results from the 2024 challenge, has introduced a new implementation of RML-LV to make it compliant with this module, along with additional changes needed for the newer test in RML-Core.

For RML-IO, BURP failed the same test cases as CARML and SDM-RDFizer. BURP generates the natural datatypes for literals in the test cases that omitted them. BURP also fails in additional test cases due to the unavailability of UTF-16 encoding and issues with datatype conversions from SPARQL.

MappingWeaver successfully executed all 41 RML-LV test cases with the usage of RMLViewer as a preprocessor. For RML-Core, test cases regarding constant-valued joins and ambiguity with regard to the generation of an invalid IRI, for which the underlying Apache Jena library throws an error, since the desired IRI, as described by the RML mapping, is invalid in the context of RDF terms. For RML-IO, MappingWeaver does not support data compression or logical target definitions in the input RML mappings, resulting in it passing only 21 of 73 test cases. MappingWeaver fails one test case for conditional operations in the RML-FNML module, and also fails three test cases requiring the usage of the *toUpperCaseURL* third-party function due to an incorrect implementation in the original library. Lastly, RML-CC is unsupported due to substantial changes required for the underlying algebra formalization for data aggregation operations.

Typhon-RML presented decent coverage of the RML-Core and RML-IO modules, with 66 out of 76 and 63 out of 73 test cases passed, respectively. Failures in RML-Core are mainly due to differences in encoding and escaping of specific characters, intended misalignment with expected behaviours (e.g.,

---

[30]https://github.com/kg-construct/rml-io/issues/151

handling of cartesian products or missing inputs), and incomplete support for recent changes in the RML specification (e.g., validation of mapping rules against not updated SHACL shapes). In RML-IO, limitations are primarily related to engineering constraints of the execution stack, including support for UTF-8 encoding only, limited support for some compression formats (e.g., "xz") due to missing Apache Camel components, and the absence of default values for null references. For RML-LV, Typhon-RML reports no results for 2026. Although the authors initially developed an experimental implementation for the KGCW 2025 challenge, supporting nested expression fields and index/record keys would require a major refactor that would significantly increase engine complexity and harm the readability of the generated MTL mapping rules. Therefore, it was decided not to merge that code into Typhon-RML and to rethink the approach in future work.

## 3.2. Track 2: Performance

A uniform hardware environment and a standardized benchmarking platform are provided for all participants to ensure fair comparison of performance. Each participant is provided with their own virtual machine with identical hardware resources, ensuring a consistent evaluation environment for all mapping implementations. Each virtual machine has the following hardware specifications: i) Intel Xeon Gold 5220 CPU at 2.20 GHz ii) 120 GiB storage space, and iii) 16 GiB RAM. Participants are required to use a modified version of the KROWN [19] benchmark specifically made for the challenge[31]. KROWN isolates the mapping engine's execution in a Docker[32] container, thereby standardizing the metrics collection process across all mapping engines. Moreover, KROWN provides a data generation module to enable reproducible scenario setups for evaluating the mapping engines.

### 3.2.1. Scenarios

We devised 6 scenarios to evaluate the performance of the participating implementations. Each scenario has its own sub-scenarios, in which the scenario-specific parameters are adjusted to simulate different workloads.

**Data-size**: The performance of the mapping engines can vary depending on the type of input data sizes (i.e., cell size, number of properties per data record). Thus, we scale input data in terms of the *number of data records*, and the *number of properties* inside each record.

**Joins**: We evaluate the engine's ability to handle join complexity by varying the *join cardinality* and the *percentage of the input data that can be joined*.

**Mappings**: Semantically similar but syntactically different RML mappings may have a negative impact on the performance of mapping engines that are not aware of such differences (i.e., two RML mappings that produce the same RDF KG but are written differently). Thus, to simulate such scenarios, we adjust the *number of triples maps*, and the *number of predicate-object maps* for each triples map.

**Named Graphs**: Premature optimizations can negatively impact performance, depending on the underlying implementations, when generating RDF Quads with named graphs [19]. This scenario adjust the *number of graph maps* definitions, and the *location of the graph maps* (i.e., graph maps defined under subject map or predicate-object map) in the input RML mappings.

**Duplicates-empty**: Data duplication and null-like records can affect the mapping engine's performance, depending on its implementation. We scale the *duplication percentages* of the input data and the *percentages of the input data that are null* for this scenario.

---

[31]https://github.com/kg-construct/kgc-challenge/tree/main/track_2_performance
[32]https://www.docker.com/

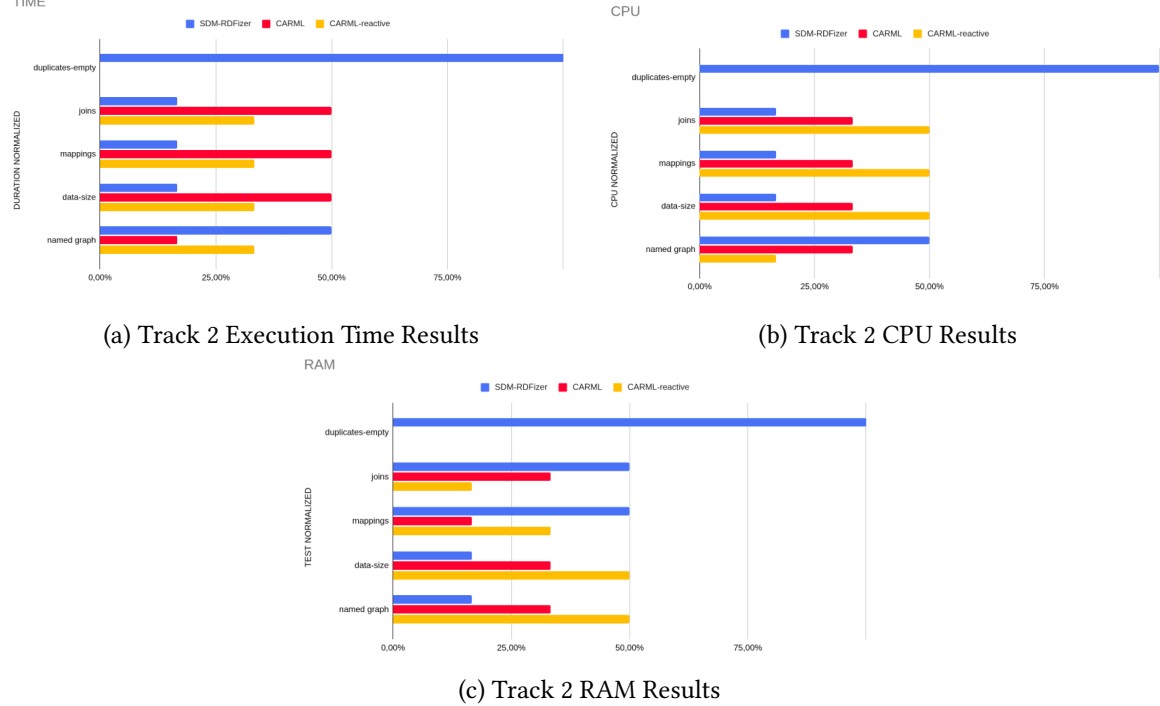

(a) Track 2 Execution Time Results

(b) Track 2 CPU Results

(c) Track 2 RAM Results

**Figure 1:** Comparative results for the performance track, where the x-axis represents the relative standing of the engines, with higher percentages indicating better performance or more efficient resource utilization.

**Collections and Containers**: RML-CC modules require aggregation of the input data to generate RDF collections and containers. Depending on the aggregation strategy of the implementation, the performance of the engine can vary. We imported the sub-scenario parameter scalings from Debruyne et al. [20] and integrated them into KROWN.

### 3.2.2. Metrics

For the comparative analysis, we measured the following metrics at fixed intervals throughout the runs: i) Execution time (ms), ii) RAM usage (GiB), and iii) CPU usage (ms). For each scenario, we use the mean value of each metric to rank the engines according to their performance for the metric. The aggregated results for these metrics and scenarios are presented in Figure 1.

### 3.2.3. Discussion

For Track 2, SDM-RDFizer, CARML, and CARML-reactive are the only participating engines for the performance evaluation. All three participants submitted results for the scenarios on Joins, Mappings, Data Size, and Named Graphs. CARML and CARML-reactive have no results for the Duplicates-Empty scenario; thus, they are excluded from the measurement for that scenario in Figure 1.

In terms of execution time (Figure 1a) and CPU usage (Figure 1b), CARML, and CARML-reactive are consistently better than SDM-RDFizer for 3 out of 4 scenarios for which results from all three engines are available. For the Named Graph scenario, SDM-RDFizer is the fastest among the three engines but has the highest CPU usage.

As shown in Figure 1c, the memory consumption of SDM-RDFizer increases with the scale of the input data. This behavior is primarily caused by the duplicate removal process. SDM-RDFizer stores every generated triple to determine whether newly generated triples are unique. Unique triples are retained and added to the KG, whereas duplicate triples are discarded. Consequently, larger inputs with lower duplicate rates yield more unique triples, resulting in higher memory consumption.

In contrast, SDM-RDFizer exhibits lower memory usage when processing data sources with high duplicate rates or large numbers of empty values. In these cases, fewer unique triples are generated, reducing the number of triples to store for duplicate detection.

Join execution also contributes to memory consumption. SDM-RDFizer stores join results to avoid redundant join computations, improving execution efficiency at the cost of additional memory usage. The amount of memory required depends on the size and selectivity of the join results. For example, joins between large data sources with high selectivity generate many matching results, increasing the memory required to store them in the PJTT data structure.

In terms of memory (Figure 1c), peak resident memory for both CARML and CARML-reactive is broadly insensitive to input scale: across the four scenarios, both variants sit between roughly 3.0 and 3.9 GB, including on the 10 million-row Data-Size cell. Both the JVM heap and CARML's in-process database buffer pool were configured to 1 GB, chosen to validate behavior under a memory-constrained budget rather than scale to the VM's available RAM; this also keeps the spill-to-disk path active on all but the smallest cells. Above the heap, CARML's additional resident memory is the buffer pool (spilling to disk when the working set exceeds it), while CARML-reactive does not stage sources into the database and instead runs the read and term-generation pipeline on the JVM heap, routing only large joins through the same spill-capable executor. The reactive heap is therefore slightly more loaded on Joins, where source-side work dominates, putting its peak just above CARML's in that scenario, while remaining lower on the others.

In the Collections and Containers scenario, only SDM-RDFizer yields results for all parametric evaluations. In contrast, CARML and CARML-reactive only produced results for sub-scenarios that included the number of records and fixed-size lists of values, with duplicates within those lists scaled according to the setups described by Debruyne et al. [20]. For CARML and CARML-reactive, the other cases were constrained by the benchmark VM's available disk: the highest-volume cells produce several gigabytes of N-Triples per run, and across the five required runs, their accumulated output can exceed the free disk space, which is mitigated by reclaiming each run's output before starting the next. For the cases of scaling join or duplicates in the lists, they are stopped in advance because a single run's output would not fit on the benchmark VM. These cases use the RML-CC gather construct to collect a cartesian product (the many-to-many join of a million-row source, or the product of two value arrays per record) into an RDF list; since every list member additionally expands into an rdf:first/rdf:rest chain, the output grows quadratically with the input. The limitation is output volume rather than the engine: CARML streams these RML-CC gathers into bounded memory, so, given sufficient disk space and time, it would, in principle, complete the sub-scenarios.

### 3.3. Track 3: Mapping Methodology

The authors of Typhon-RML also participated in Track 3. They conducted an exploratory study to investigate the use of an agentic approach for automatically generating mapping rules in MTL. The experiments showed that obtaining a working solution for the proposed scenarios in Track 3 often requires multiple iterations, and the authors were unable to finalize a submission because the end-to-end experimentation time was longer than expected due to the inherent loop of agentic reasoning and repeated execution/validation cycles. While the agent can eventually produce mapping rules that satisfy given input/output pairs, the exercise highlighted open issues on how such solutions should be properly evaluated in a challenge setting. Beyond functional correctness, mapping approaches (e.g., mapping language used) should be taken into account, and mapping quality also involves readability and generalizability to unseen examples. For instance, some generated rules may overfit specific pairs or introduce custom functions whose necessity is unclear, raising the need for explicit evaluation criteria and metrics (potentially including human judgment). Moreover, the cost and comparability of different approaches are difficult to measure: unlike a single prompt, an agent may span many iterations (with unpredictable costs, e.g., in tokens), benefit from user-provided hints over time, or rely on dynamically defined scripts. A potential recommendation for the next edition of the challenge is to define more specific tasks for Track 3 (e.g., automatic conceptual mapping vs mapping rules definition using different

mapping approaches).

## 4. Conclusions

The three tracks of 2026 KGCW Challenge provide not only a comparative analysis of the participating mapping engines, but also insights and improvements to the RML specification and toolings for the benefit of the wider KG construction community.

The results from Track 1 on latest RML specification show that majority of the mapping engines (3 out of four) are almost compliant fully compliant with the five modules of RML, namely, RML-Core, RML-IO, RML-CC, RML-FNML, and RML-LV. We publish an updated implementation report[33] on the conformance status of the participating engines using the results from Track 1 on RML conformance. This report will be updated in the future as more implementations are submitted. Moreover, because the participants rigorously validated both their implementations and the correctness of the test cases, we have received patches to fix the test cases for the modules RML-Core, RML-IO, and RML-FNML.

Although there are only two participants for Track 2 on performance (CARML, and SDM-RDFizer), the organization of this track provides us with improvements to the KROWN benchmarking software with new scenarios using new modules such as RML-CC. Furthermore, the issues reported by participants regarding bugs in the benchmark software will be used to improve KROWN.

Finally, the valuable feedback by the authors of Typhon-RML for Track 3 on mapping methodology will be used to refine the scope of the track in future editions of the challenge. The revised track will define clear objectives and specific tasks for participants to complete. It will also include a concise set of qualitative evaluation criteria to facilitate the comparison of different approaches to knowledge graph construction.

## Acknowledgments

Enrique Iglesias and Maria-Esther Vidal are supported by the "Leibniz Best Minds: Programme for Women Professors", through funding of the "TrustKG-Transforming Data in Trustable Insights" project (Grant P99/2020), and by the Lower Saxony Ministry of Science and Culture (MWK) with funds from the Volkswagen Foundation's zukunft.niedersachsen program (CAIMed - Lower Saxony Center for AI and Causal Methods in Medicine; GA No. ZN4257). Sitt Min Oo is supported by the imec.icon project PACSOI (HBC.2023.0752), which was co-financed by imec and VLAIO and brings together the following partners: FAQIR Foundation, FAQIR Institute, MoveUP, Byteflies, AContrario, and Ghent University – IDLab. Jakub Duchateau is supported by the Fonds de la Recherche Scientifique – FNRS under Grant n° MIS F.4016.24. The work done by Marco Grassi and Mario Scrocca has been partially funded by the European Commission through the Chips Act Joint Undertaking project SMARTY (Grant no. 101140087). Adrián Martínez-Balea is funded by the Agencia Estatal de Investigación (Spain) (PID2023-149549NB-I00), the Xunta de Galicia — Consellería de Cultura, Educación, Formación Profesional e Universidades (Centro de investigación de Galicia accreditation 2024–2027 ED431G-2023/04) and the European Union (European Regional Development Fund — ERDF).

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
