# OpenReview forum: "Technical Report on the 4th Edition of Knowledge Graph Construction Workshop Challenge"
_eswc-conferences.org/ESWC/2026/Workshop/KGCW — ESWC 2026 Workshop KGCW Submission_

### Official Review · ~Franck_Michel1 · 2026-06-09
**Clear and useful report**

**Rating:** 7
**Confidence:** 5

**Review:**

This paper is a report of the Knowledge Graph Construction Workshop Challenge 2026.
It describes the participants in the challenge, the challenge tracks and scenarios, the metrics used to compare the candidates, and provides the results.

Overall the report is clearly written and provides the necessary inputs to comprehend the goals and results of the challenge.
My only suggestion for a minor change would be this: the last paragraphs of sections 2.2 and 2.3 suggest future works of the SDM-RDFizer and BRUP. They could probably move to section 4: the conclusion ends with how the feedback from participants could help improve the challenge, and conversely it could provide future works that the authors figured out while participating in the challenge.

Below are a set of minor details, typos and suggestions of rewording.

--

Section 2.1 lacks a few links e.g. “Project Reactor stream operators“, DuckDB.

Section 2.4 mentions RML v1.1.2 and url https://rml.io/specs/rml/  in footnote. It is actually the first time I hear about this versioning. And I can’t see it either on the Github releases. This reference should be clarified (link to the specific version) or removed.

In 2.5, a misplaced sentence gives challenge results, whereas this is not the case for the others for which results are left to section 3.

Table 1 is provided in section 3 but not referenced in the text.

In Figure 1, the labels are very tiny and hardly readable.

Typos and wording issues:

-	"Previous editions of the challenge have focused ON evaluating mapping engines based on RML and its ecosystemS"

-	"we organize the 4th Edition of KGCW Challenge where each participants submit their implementations": remove "each"

-	“For this paper, we organize the 4th Edition of KGCW Challenge”: not sure what you want to say, e.g. : In this paper, we report the results of the 4th Edition of KGCW Challenge that we organized, …” ?

-	About CARML: “Every mapping is resolved into a graph of logical views before execution”: the wording “resolved into a graph…” sounds somewhat awkward.

-	The limitation is THE output volume rather than the engine

-	“With BURP-Error, they did some refactoring in BURP. We replace the Jena RDF model …”: mix-up of subjects and tenses: “they” and “we” = the authors.  Use of past in first sentence, and present in the next sentence.

-	“SDM-RDFizer is publicly available on GitHub” : more importantly it is available under an open license.

-	Footnote 20 give URL https://agentskills.io/. Was it the goal or should it give the link the specific skill of Typhon-RML?

-	This scenario adjusts the number of graph maps definitions

-	4. “The results from Track 1 on latest RML specification show that majority of…” : the majority of

-	“are almost compliant fully compliant”: duplicated “compliant”

In the bibliography, uppercases are not respected in the titles.

---

### Official Review · ~Michael_Freund1 · 2026-06-10
**Clear engine overview and challenge summary**

**Rating:** 7
**Confidence:** 5

**Review:**

## Overview
The technical report presents the results of the 4th Knowledge Graph Construction Challenge, co-located with the 7th International Workshop on Knowledge Graph Construction. The challenge evaluates KG construction engines on (i) conformance with the updated RML specification, (ii) performance in terms of execution time and memory consumption, and (iii) alternative mapping methodologies.

The report first gives a nice overview of the participating engines and highlights their current capabilities and limitations. It then presents and discusses the results of the challenge.

## Strenghts
- S1: The report provides a clear overview of the participating engines and explains what distinguishes them technically.
- S2: The conformance results are presented clearly, especially through the summary table.
- S3: The discussion of problematic test cases is useful and increases the transparency of the evaluation.

## Weaknesses
- W1: The Track 2 diagrams are quite small and hard to read. Larger diagrams would improve readability.
- W2: A section summarizing the overall state of the evaluated engines, including missing features, usability aspects, and future directions, could help readers decide which engine to use. Additionally, it could be useful to discuss how to use the engines in practice, for example via a CLI, configuration files, a GUI, or as a library.

---

### Decision · Program_Chairs · 2026-06-17

**Decision:**

Accept

**Comment:**

The paper is accepted for the proceedings. Please, follow the instructions of the CR and consider the comments from the reviewers: https://w3id.org/kg-construct/workshop/2026/instruction-cr.pdf

Deadline for the CR: Monday 22nd of June.